# Neutrophil Extracellular Traps: Current Perspectives in the Eye

**DOI:** 10.3390/cells8090979

**Published:** 2019-08-27

**Authors:** Gibrán Alejandro Estúa-Acosta, Rocío Zamora-Ortiz, Beatriz Buentello-Volante, Mariana García-Mejía, Yonathan Garfias

**Affiliations:** 1Research Unit, Cell and Tissue Biology, Institute of Ophthalmology Conde de Valenciana, Mexico City 06800, Mexico; 2Department of Biochemistry, Faculty of Medicine, Universidad Nacional Autónoma de México, Mexico City 04510, Mexico

**Keywords:** neutrophils extracellular traps, ophthalmology, diseases

## Abstract

Neutrophil extracellular traps (NETs) have been the subject of research in the field of innate immunity since their first description more than a decade ago. Neutrophils are the first cells recruited at sites of inflammation, where they perform their specific functions, including the release of NETs, which consist of web-like structures composed of granule proteins bound to decondensed chromatin fibres. This process has aroused interest, as it contributes to understanding how pathogenic microorganisms are contained, but they are also associated with pathophysiological processes of a wide range of diseases. Currently, there are growing reports of new molecules involved in the formation and release of NETs. However, whether the release of NETs contributes to eye diseases remains unclear. For this reason, the overall aim of this review is to gather current data of recent research in the ophthalmology field, where there is still much to discover.

## 1. Introduction

Neutrophils are the main effector cells of an acute inflammation. Their anti-inflammatory role is due to their specialised functions [1]. More than a decade ago, it was posited that, upon activation, neutrophils release extracellular structures composed of granular and nuclear constituents that neutralise bacteria. Since then, these fibrous networks have been called neutrophils extracellular traps (NETs) [2]. As this phenomenon was originally considered a particular form of cell death, different from necrosis or apoptosis, the process was labelled “NETosis” [3]. Recently, this concept has changed due to reports of two forms of NETosis: suicidal and vital [4]. The controversy continues as to whether a NETs release is a physiological host defence process or whether it is a consequence of cellular rupture [5]. It is important to be aware that robust effector functions may also lead to tissue damage [6]. The function of NETs is to offer a physical barrier that foils infectious spreading and raise the extracellular compartment of the concentration of antimicrobial substances [7]. Current research of NETs is expanding worldwide, and new research has uncovered new pathways in terms of understanding the mechanism for activating and releasing NETs. In the present review, we discuss the role of NETs in pathological responses, focusing specifically on eye diseases.

## 2. Structure of Neutrophil Extracellular Traps

Although the overall composition of NETs remains unknown, it has been described that the main components of NETs structures are nuclear DNA and histones. Also, proteins contained in different neutrophil granules like bactericidal enzymes, such as neutrophil elastase (NE), myeloperoxidase (MPO), lactotransferrin (LTF), gelatinase, cathepsin G (CG), leukocyte proteinase 3 (PR3), calprotectin, cathelicidins, defensins, and actin, among many others, conform NETs [8].

NET scaffolds are composed of DNA fibres that are 15–17 nm in diameter and globular domains with diameters of 25–50 nm [9]. NETs can be studied using different techniques. In recent years, several strategies have been developed to identify NETs in vitro. Fluorescent microscopy is a versatile method that utilises different fluorescent-conjugated antibodies to specifically detect proteins, in combination with cell-membrane selective DNA-intercalating probes. Although there are some software programs that attempt to analyse NETs, but this technique is still observer-dependent, which is its main limitation. [10]. Another technique to study the release of NETs combines fluorescent microscopy and flow cytometry, also called multispectral imaging flow cytometry (MIFC) [11]. Scanning electron microscopy (SEM) has been useful to identify NETs. However, remnants of fibrin, fibronectin, and collagen can mimic NETs, revealing that SEM has limitations when NETs are analysed in tissue [12]. In order to quantify NETs, several methods have been described [11,13,14,15,16]. Each method offers alternatives for quantifying the release of NETs. So far, however, there is no single method for quantifying NETs.

## 3. Releasing of Neutrophil Extracellular Traps

The first description of NETs was performed with several stimuli, such as phorbol 12-myristate 13-acetate (PMA), bacterial lipopolysaccharide (LPS), and interleukin-8 (IL-8) [4,17]. Since then, a substantial number of inducers have been described, including harmful agents such as bacteria, fungi, protozoa and viruses, as well as their components [18]. In recent years, an increased number of activators have been proposed to induce the release of NETs, such as high-glucose medium, complement-derived peptides, autoantibodies, cigarette smoke, urate crystals, calcium ionophore, and activated platelets [4,19,20,21]. The mechanism by which neutrophils release NETs is not completely understood. It seems that the NETs releasing pathway depends on the stimuli. Recently, de Bont and colleagues reported that LPS, PMA, and the calcium ionophore A23187 promote NETs releasing through different pathways [22]. Another interesting study has mentioned that diverse stimuli for releasing NETs are heterogeneous in terms of both protein composition and post-translational modifications by means of proteomic approaches. This suggests that NETs induced in different conditions may have different biological effects [23]. 

## 4. Suicidal and Vital NETosis

Suicidal NETosis is the first and most studied model. Several stimuli can promote this process, but the best studied are those stimulated through PMA. In this context, it should be noted that this mechanism is dependent on reactive oxygen species (ROS) generation [4,5,24]. It persists for 2–4 h and consists of a specific neutrophil lysis, which begins with chromatin decondensation, followed by nuclear membrane disintegration and a mixture of DNA material and cytoplasmic components, including neutrophil granule contents. This leads to plasmatic membrane dissolution and cell death [24]. It has been reported that there are two different mechanisms of vital NETosis, depending on the origin of DNA release—nuclear or mitochondrial DNA. The former is a ROS-independent generation process; this mechanism has been described as occurring with staphylococcus aureus stimulation, and it has been observed that NETs release is through a vesicular-dependent manner and is very rapid (5–60 min) [25]. Meanwhile, in the latter process, NETs that are released contain only mitochondrial DNA [26]. In this non-cell death mechanism, the NETs release is preserved, as well as the phagocyte neutrophil capability. This seems to represent a response against microorganism aggression [6,26]. 

## 5. Neutrophil Extracellular Traps Pathways and Involved Molecules

The mechanism by which NETs are released is still controversial and is being studied by various research groups. Since their first description to date, we know that more than 30 components of the neutrophil granules are involved, and fibres are composed of DNA and histones. Thus, NETosis processes have been described as aforementioned. We also know that, on one hand, they contribute to inflammatory resolution processes, but on the other, they can contribute within the pathophysiological mechanism of diseases, although the exact molecular and biochemical mechanisms involved in NETs release are not completely understood. 

The NETosis process has been described by several authors as dynamic, beginning with the disappearance of the nuclear structure and followed by a rupture of the nuclear envelope. Then, the components of the nucleus portion are commingled with the cytoplasm. Finally, the rupture of the neutrophil membrane occurs, although we know that the latter does not occur with certain stimuli. Independent of the stimulus, the NET release process is independent of transcription [27]. The release of NETs is evolutionarily conserved within the animal and plant kingdoms and has been described in different species [4]. 

It has been described that suicidal NETosis model depends on the protein kinase C activity (PCK) as the key modulator of this pathway, which is also NADPH oxidase 2 (Nox 2) dependent [5]. This pathway initiates with the recognition of any stimulus that activates receptors, which induces the activation of signal-related kinases, such as PKC, p38, PI3K, Src, Raf/MERK/Erk, Akt, and Ark [28,29,30,31]. All of them are well-known and important kinases for Nox2-dependent NETs formation [30,32]. Further downstream, the multimeric complex of NADPH oxidase assembles at the phagosomal membrane and generates ROS. With ROS generation, cytosolic calcium increases, acting as a cofactor for peptidyl arginase deaminase 4 (PAD4). PAD4 is an enzyme that endorses deamination of histones, and therefore, favours chromatin decondensation [24]. However, there is evidence suggesting that physiological stimuli induce PAD4-dependent ROS-independent NETosis [33]. 

At the same time, while neutrophil elastase (NE) and myeloperoxidase (MPO) translocate from cytoplasm to the nucleus [34,35], MPO binds to chromatin and synergises with NE decondensing chromatin [34]. Subsequently, Nox2 complex produces ROS that work as second messengers, promoting nuclear membrane disintegration. This oxidative activation is important because NE binds to F- actin filaments in the cytoplasm and must degrade them in order to enter the nucleus [35]. This process allows nuclear content mixtures into the cytoplasm, ultimately loosing the plasmatic membrane and then discharging the neutrophil contents as extracellular traps [6,24,36]. Interestingly, pharmacological inhibition of NADPH oxidase or ROS scavengers block NETs formation with certain stimuli [32]. Moreover, neutrophils from patients’ chronic granulomatous disease, a primary immunodeficiency with mutations in NADPH oxidase subunits, lose their ability to release NETs in response to mitogens and some microbes, genetically confirming the relevance of ROS [37]. In contrast, there are some NETs stimuli, such as immune complexes, ionomycin and nicotine, that have been proposed to trigger NETosis independently of NADPH oxidase, depending instead on mitochondrial ROS [38].

Vital NETosis, in contrast to suicidal NETosis, is a live cell process at the time of releasing NETs. Neutrophils can release part of or the entire nucleus without breaching the cell membrane, resulting in an anuclear cytoplasm that is still able to move and phagocytose bacteria [4]. Thus, in this type of NETosis, neutrophils preserve some of their functions. 

In this process, several molecules are involved, including MPO and NE. PAD4 is a nuclear enzyme that performs the citrullination of histones through the change of arginine for citrulline, making the overall charge of the histones less positive, particularly in histone H3 (H3Cit) [39]. This results in a lower affinity for the negatively charged DNA, thereby stimulating chromatin decondensation and eventually releasing NETs [22]. Although it has been described that PAD4 activation concludes in DNA decondensation, NE is sufficient for nuclei decondensation in vitro. However, the mechanism that disassembles the nuclear envelope in neutrophils is still unknown [34].

The mechanisms that regulate PAD4 activation in the NETs releasing process are not yet fully understood. There are still many controversies about the role of PAD4 activity in the release of NETs. For example, it has been described that when calcium ionophore is used, PAD4 is activated and a release of NETs occurs [22,38,40]. In contrast, neutrophil activation with PMA is a potent signal for releasing NETs in a PAD4-independent manner [41,42,43]. Moreover, it has recently been demonstrated that PAD4 activity is necessary to NET formation in a bacterial presence [44]. 

In this regard, Konig and Andrade have described two mechanisms that are different from NETosis: leukotoxic hypercitrullination (LTH) and defective mitophagy. Both have been erroneously classified as “NETosis” [40]. These authors showed evidence that PAD4 activation is independent of PMA. Actually, when PMA is used as an activator, all protein citrullination including H3Cit is abolished. Moreover, LTH is a NADPH oxidase-independent phenomenon and not necessarily bactericidal; in this process, NET-like structures (NLS) are released [40]. There are several features that distinguish LTH from NETosis—(1) NLS are triggered by prominent and sustained calcium influx and are inhibited by chelation of extracellular calcium [45], (2) NLS are generated independently of NADPH oxidase activity [45], (3) NLS require PAD4 activity and are suppressed by PAD inhibitors [41,46], (4) NLS undergo rapid formation (within minutes) [41], and (5) in LTH, protein citrullination is not limited to histones and transcription factors but encompasses proteins across all molecular weights [47]. On the other hand, defective mitophagy has not been associated with protein citrullination [40,48]. This has been explained as a compensatory mechanism for defect mithophagy. This mechanism is enhanced in neutrophils by inflammatory signals such LPS or C5a after priming with GM-CSF [26].

### NETs and Disease

Despite NETs possessing antimicrobial activities and aiding in the resolution of inflammation [43], they also are pathologic in multiple diseases, such as autoimmune diseases [49], including arthritis [50,51], systemic lupus erythematosus [52,53,54,55], antiphospholipid antibody syndrome [56,57], small vessel vasculitis [58], and psoriasis [59,60,61,62]. In cancer, neutrophils affect health through multiple mechanisms, and evidence for the role of NETs have been found [63,64,65,66,67,68,69] and continue to emerge [38]. In the cardiovascular system, NETs have played a role in atherothrombosis and venous thrombosis [70,71,72]. Another important disease in which NETs have relevance in its pathophysiology is sepsis. In this process, it has been demonstrated that a release of NETs increases the risk of venous thromboembolism (VTE). It has also been shown that a high percentage of NETs correlates with a high risk to present sepsis [73]. Moreover, the presence of NETs has induced multiple organ damage in an experimental sepsis model [74]. In lung diseases, NETs have an important role in chronic inflammation, including cystic fibrosis [75,76] and chronic obstructive pulmonary disease (COPD) [21,77,78,79].

## 6. Eye Diseases 

As we have discussed earlier, the release of NETs is an aspect of damage seen in a wide variety of diseases. For a long time, it was considered that the eye was an immune-privilege organ without any immune response. Nevertheless, this idea is not entirely true. Several works have described that ocular immune privilege provides the eye with immune protection against inflammation in order to minimise the risk to vision [80], and the focus was on the ocular privilege associated with Treg response. On the other hand, we know that different diseases have an immune response, and the eye has a series of mechanisms that defend against infections, but it is also true that there are pathologies that are not yet known to regulate this. 

In recent years, there has been evidence of NETs and their implication within pathophysiology in ocular diseases. 

## 7. Cornea and Ocular Surface 

The cornea is the transparent and avascular dome-shaped tissue of the eye. It represents a physical immune barrier, and together with the tear film, provides the anterior ocular refractive surface. Its transparency is due to the composition and physiology of its cellular constituents [81].

Eye rheum is a medial angle accumulated discharge, independent of any disease. Its composition consists mainly of leukocytes, neutrophils, and their related proteins. Aggregated neutrophil extracellular traps (aggNETs) have also been measured in the eye rheum, and it has been concluded that their presence prevents the spreading of inflammation because they degrade inflammatory mediators [82,83]. However, it has been found that chronic accumulation of aggNETs contributes to inflammation and tissue damage. 

### 7.1. Dry Eye Disease

Dry eye disease (DED) is defined as a multifactorial disease of the ocular surface, characterised by a loss of homeostasis of the tear film and accompanied by ocular symptoms, in which tear film instability and hyperosmolarity occurs. Ocular surface inflammation, damage and neurosensory abnormalities play etiological roles [84]. The dynamic turnover of the corneal epithelium is increased in DED. This is regulated by apoptosis, which stimulates the immune system, leading to the extracellular DNA (eDNA) release and NETosis. It has been suggested that, in healthy eyes, the eDNA is cleared by nucleases contained in tears, but in patients with DED (mainly in severe cases), it has been found to be nuclease deficient. This permits the tear gathering of eDNA and NETs to trigger and perpetrate inflammation of the ocular surface [85]. This theory was supported by Tibrewal and colleagues, who reported the presence of excessive amounts of eDNA in the tear fluid of patients with DED associated with a release of NETs, which was higher in patients with worse DED, which in turn is associated with hyperosmolar stress and the deprivation of nucleases [86]. Therefore, a high amount of NETs is expected to be found on the ocular surface in patients with DED. The outcomes of these studies were supported by Mun and colleagues, through a pilot clinical trial of recombinant human deoxyribonuclease I (0.1% DNase) eye drops in patients with DED. After comparing the use of DNase versus placebo eye drops in 41 patients four times a day for eight weeks, they demonstrated a significant reduction in symptomatic and clinical severity of DED. In relation to the mechanisms previously described, this result could possibly be explained due to degradation of the NETs [87]. 

Ocular sicca in chronic graft-versus-host disease (GVHD) has also been related to NETosis. GVHD is a complication that takes place in patients after hematopoietic stem cell transplantation, and ocular GVHD (oGVHD) is a particular clinical manifestation of chronic GVHD. Clinical signs and symptoms of oGVHD are related to eyelid disease and tear deficiency. However, it is not considered merely as a specific form of tear deficiency or dry eye. In fact, oGVHD has different pathophysiology and treatment than DED, but it is associated with a dysregulated immunity response. Through the analysis of ocular surface washing from patients with oGVHD, An and colleagues demonstrated an increase in associated NET-cytokines levels, such as MPO and IL-8, among others, which are well-known to be chronic inflammation contributors. Also, they showed therapeutic effects of sub-anticoagulant dose heparin (100 IU/mL) eye drops through the destabilisation and clearance of NETs from the ocular surface [88], demonstrating the essential role of NETs in oGVHD.

### 7.2. Infectious Keratitis

The leading cause of monocular blindness worldwide is corneal disease, and infectious keratitis is among the leading causes of corneal opacities. This pathology is more common among marginalised populations. The prognosis and clinical outcome of this ocular pathology is determined by the proper identification of the causative microorganism, as well as effective management. 

In comparison with other tissues, the late immune response against infectious microbes could be significant because of corneal avascularity. The epidemiology of infectious keratitis is related to the population studied. While in developed countries it is associated with bacterial ulcers, mainly due to contact lens use, in developing countries, fungi are common causative agents [89]. Several keratitis cases in contact lens wearers are often caused by *Pseudomonas aeruginosa*, a Gramm negative bacterium with a harmful evolution that can cause permanent vision loss. The immune reaction against *Pseudomonas* is dominated by neutrophil response, but the infiltration pattern changes upon the *Pseudomonas* strain [90,91]. When Shan and colleagues tested the ability of *Pseudomonas* strains to release NETs and their vulnerability to be NET-captured, they found that even cytotoxic strains are better inductors of NETs and are less sensitive to be NET-captured [90]. Since the first description of NETs made by Brinkmann in 2004, the antimicrobial properties of NETs were described and explained through degradation of virulence factors and bacteria killing [2]. Current data has explained some NET evasion mechanisms in keratitis are caused by *Pseudomonas* strains. *Pseudomonas* employs virulence factors, like the type-3 secretion system (T3SS). T3SS also forms biofilms through Psl exopolysaccharide and favours bacterium NET escape. The inefficacy of neutrophils to penetrate *Pseudomonas* biofilm leads to the production and release of NETs as an attempt to avoid bacteria spreading [92]. The shedding of outer membrane vesicles (OMVs) contributes to avoid *Pseudomonas* linking to NETs; thus, the shedding of OMVs as a possible therapeutic target should be considered [93]. 

Mycotic keratitis is usually developed from corneal injury due to agricultural work or, less frequently, in contact lens users. Ulcers caused by fungi have worse outcomes than bacterial ulcers [93]. Although neutrophils can clear conidia fungi forms by phagocytosis, the hyphae of fungi are too large to be cleared by this mechanism, and NET release has been reported as a capture and kill neutrophil machinery in yeast and hyphal forms. Jin and colleagues studied the relationship between the quantification of NETs and the prognosis of fungal keratitis in vivo. They studied and measured DNA-releasing neutrophils in 14 patients with clinical and final diagnostic biopsy of fungal keratitis, in which the culture was positive for *Fusarium* sp., *Aspergylus* sp., *Candida*, and *Alternaria* sp. They demonstrated the existence of NETs in different stages from day 2 to day 22, suggesting their role in the entire stage of this infectious pathology, but they did not find any relationship between the number of NETs and the size of the ulcer. However, they did identify a close relationship between the patient response to therapy and the quantity of NETs [93]. 

## 8. Corneal Injuries and Repair 

Due to its location, the cornea is susceptible to possible injuries like abrasions, burns, infections, and de-epithelisation. Depending on the damage, those stimuli can trigger the corneal wound healing process, impacting on the tissue transparency and, in some cases, producing permanent impairment of vision [94]. As previously studied, the exacerbated production of NETs offers the basis for the progress and preservation of inflammation. We have previously reported that NET release is presented in a rabbit corneal alkali burn, and intracameral injection of human amniotic mesenchymal stem cells (hAM-MSC) was able to significantly inhibit NET release. hAM-MSC were also able to reduce the number of inflammatory cells and infiltrated neutrophil, as well as neovascularisation and corneal opacity. These effects are attributable to their immunosuppressive molecules [95]. 

## 9. Uveitis 

Uveitis refers to a group of eye diseases that are defined by intraocular inflammation, specifically affecting the uveal tract, which compromises the iris, ciliary body, and choroid [96]. It also includes other inflammation of adjacent intraocular structures, such as the retina, vitreous, and optic nerve and, depending on the severity, can generate imminently harmful ocular injuries [97]. Uveitis can be acute, chronic, or recurrent, infectious or non-infectious, granulomatous or non-granulomatous, and unilateral or bilateral [98,99]. 

The main non-infectious causes of uveitis are acute anterior uveitis, Behçet’s disease (BD), Vogt-Koyanagi-Harada (VKH), and juvenile idiopathic arthritis (JIA), among others. Interestingly, it has been described that in non-infectious uveitis, the immune response is the main responsible cause [100].

The main infectious aetiologies of uveitis include pathogens like the herpes virus, *Toxoplasma gondii*, *Mycobacterium tuberculosis*, and *Treponema pallidum* [99], among others. To date, it is estimated that uveitis is responsible for about 10% of legal blindness in the United States per year. The socioeconomic impact of uveitis lies in the main affected age group being the young and middle working-age population; thus, the patients are usually economically active [101,102].

BD is a systemic vasculitis that affects arteries or veins [103]. Clinical features consist of recurrent oral and genital ulcers associated with inflammatory manifestations with skin, eyes, joints, gastrointestinal, and central nervous system involvement. It is histologically diagnosed by an intense neutrophilic infiltrate. There are reports that describe an inappropriate hyperactivity of neutrophils by cytokines and other molecules that produce increased volumes of superoxide anion via NADPH oxidase (Nox2) [103,104]. Clinical criteria for ocular BD diagnosis include anterior and posterior ocular segment involvement: uveitis, hypopyon iritis and retinitis. The wide range of ocular BD features include mild to severe clinical conditions. The most common anterior segment BD manifestation is bilateral non-granulomatous anterior uveitis, usually associated with transient hypopyon, that responds to topical steroid treatments. Posterior segment presentation includes retinitis, vitritis, and retinal vasculitis, which can provoke complications such as cystoid macular oedema and retinal ischaemia, with subsequent retinal neovascularization [105]. In a study conducted by Safi and colleagues, it was described that patients with BD had vasculitis as a result of a high release of NETs. It is also mentioned that 58% of patients had ocular manifestations, such as anterior uveitis, posterior uveitis, and panuveitis, among others [106], suggesting an active role of NETs release in BD aetiopathogenesis. In another study conducted by Perazzio and colleagues, they studied how the neutrophils from patients with severe BD showed up before and after PMA stimulation, with an increase in their oxidative burst activity via a CD40-dependent phosphoinositide 3-kinase/NF-κB pathway. This was compared with cells from patients with mild manifestations of BD [107]. This provided the basis for this team to study the effect of the soluble CD40 ligand (sCD40L), a member of the TNF family, in relation to the release of NETs. This soluble form of CD40 is related to the antigen-presenting cell process, T-cell activation, and platelet aggregation [107,108]. Current data has shown a rise of sCD40L serum levels in BD patients. Recently, Perazzio and colleagues demonstrated a significant increase of NET release and reactive oxygen species production after neutrophil stimulation with sCD40L and a decrease by blocking this molecule [107]. This indicates an active function of this molecule as a possible target for the treatment for BD or other related NET releasing pathologies. 

Although increased NET release has been studied in systemic lupus erythematosus and rheumatoid arthritis, there are no current studies of NET release and its association with ocular manifestations of these diseases [50,53]. We consider BD as ocular symptoms that are required criteria for the diagnoses. 

## 10. Vitreoretinal Pathologies 

Diabetic retinopathy (DR) is one of the main causes of avoidable bilateral blindness worldwide [109,110,111]. It has been considered as a microvascular complication related to diabetes mellitus, and its diagnosis and classification is based on visible vascular lesions on the retina. There are two main stages—non-proliferative DR (NPDR) and proliferative diabetic retinopathy (PDR). In the first stage of PDR, microaneurysms, retinal haemorrhages, and vascular tortuosity are present in the ophthalmoscopical examination, while in the second stage, frank neovascularization is found [111]. An important additional categorisation in DR is diabetic macular oedema (DME), which is a fluid accumulation into the neural retina that leads to abnormal retinal thickening and often cystoid oedema of the macula [111]. It is one of the major complications of DR [112]. The hallmark of DR pathophysiology lies in the compromised integrity of blood–retinal barrier (BRB). It has been described that long-term hyperglycaemia promotes an increase in vascular capillary permeability, letting neutrophils pass through it, infiltrating choroid and retina and therefore enabling retinopathy progression, which is caused by this chronic inflammation condition [113,114].

Within the pathophysiology of DR there are reports of pro-inflammatory molecules, as well as growth factors involved in the exacerbation of the disease. Several inflammatory cytokines, such as IL-1ß, IL-6, IL-8, tumour necrosis factor-alpha (TNF-α), and monocyte chemoattractant protein-1 (MCP-1), have been reported as elevated in vitreous samples from NPDR patients [115]. In contrast, another study of patients with PDR described elevated levels of pro inflammatory cytokines, IL-1ß, IL-6, IL8, and CCL2. However, they reported that IL-10 was similar to that obtained in the controls [116].

Some important factors such as endothelin 1 (EDN1, also called ET-1), vascular endothelial growth factor (VEGF), and TNF-α have been involved in the inflammatory reactivity and neovascularisation of PDR. By another hand, in patients with DME, it has been found that the levels of angiopoietin-2 (ANg2), an important modulator of angiogenesis of VEGF, is elevated in comparison to controls [111]. In recent years, the use of therapy based on intravitreal injection of anti-vascular endothelial growth factor (anti-VEGF) drugs has become the first line of treatment for PDR [112]. In order to describe another therapy, experiments in vitro have demonstrated that corticosteroids also modulate vascular permeability by suppressing the production of VEGF mRNA, VEGF-mediated protein expression, and VEGF receptor in human cell cultures, macrophages, and endothelial cells [112].

As aforementioned, DME is the major complication in PDR; thus, treatment alternatives have been investigated. Intravitreal steroid administration has shown to be an alternative to anti-VEGF drugs in treatment of naïve eyes affected by DME. This reduced the frequency of anti-VEGF intravitreal injections in patients with coronary diseases, for whom anti-VEGF agents are contraindicated [112,117]. 

Certain studies associated NET release as a factor involved in the pathogenesis of DR. Wang and colleagues showed that neutrophils from diabetic patients, especially in PDR, are able to promote spontaneous release of NETs and that neutrophils of healthy controls are able to release NETs when they are stimulated by high glucose levels. They also studied the association between high glucose-induced NETosis and NADPH oxidase-derived ROS pathways [113].

Although it has been studied and reported that high glucose medium can increase NET release in vivo and in vitro [118], it has even been reported that levels of NET markers are independent factors for diabetic retinopathy [119]; yet, the molecular mechanisms that stimulate its release has not been clearly elucidated. 

The existence of NETosis has been reported in ocular inflammation induced by proinflammatory cytokines in a mouse model and in samples obtained via standard pars plana vitrectomy of diabetic patients with PDR. In the same study, they confirmed the presence of NETs, and these authors proved in their mouse model that NETs can be degraded with DNase I, which has been reported to help in clinical NETs association when used as eye drops [86].

Barliya and colleagues have demonstrated the existence of NET release in a mouse model and human samples. They injected two well-known inflammatory cytokines, IL-8 and TNF-α, into murine eyes as NET inductors. NETosis was measured by considering specific marker staining for MPO, NE, and H3Cit. They found aggregate neutrophils in anterior and posterior chambers with NET staining markers. Their results confirmed eDNA through the inhibition of H3Cit staining when DNAase treatment was applied. In the same study, the authors analysed vitreous human samples of diabetic patients with proliferative retinopathy that underwent vitrectomy and correlated NET release with the severity of the disease. As expected, they reported that those patients with worse clinical features showed a higher release of NETs [120]. 

## 11. Age Macular Degeneration 

Age macular degeneration (AMD) is an acquired central retina degeneration and the leading cause of irreversible visual impairment in elderly people. Within its aetiology, environmental factors are well-described as triggers for the disease in both of its forms: dry or atrophic and exudative or neovascular “wet” AMD. The latter is less frequent but the most serious because it causes severe visual loss [121].

The pathogenesis of neovascular AMD is multifactorial; however, it is well-known that the hallmark of macular oedema is the breakdown of the blood retinal barrier (BRB), which leads to vascular leakage. This condition is common in retinal diseases, such as diabetic retinopathy, cystoid macular oedema, ischaemic retinal vein occlusion and some forms of posterior uveitis [122]. The breakdown of the BRB facilitates immune cell infiltration, as previous reports show [123]. Although the molecular mechanisms are not well-understood, it has been reported in histopathological studies of AMD eyes that the presence of “drusen”, lipoproteinaceous undigested products of retinal pigment epithelium (RPE) exist between the RPE and Bruch’s membrane, as well as basal laminar deposits, which are remarkably related to choroidal neovascularisation [124]. Histological reports of these infiltrates have proven the existence of complement molecules like C3, C5b-9, and the membrane attack complex (MAC), as well as macrophages that, due to TNF stimulation, initiate VEGF production by RPE [121,125]. The retina is susceptible to these factors and produces inflammatory cytokines such as VEGF and tissue factors that induce fibrin growth, which act as a platform for Choroidal Neovascularisation (CNV) development [121].

Furthermore, it is known that VEGF not only contributes to vascular dysfunctions, but acts as a pro-inflammatory molecule, as long as it endorses the expression of inflammatory cytokines like IL-8 [126], a biomarker of NET release, and promotes ROS production derived from NADPH oxidase in turn [115]. VEGF is also a regulator of angiogenesis and a main contributor of macular oedema development either in AMD or DR. While the molecular pathogenesis of these pathologies is different, they have a background of inflammatory microenvironment in common [111,127]. 

Although there is no current data of NETosis related to AMD, as previously mentioned, there is a study on a diabetic rat model where the existence of NETs was demonstrated in eye tissues, specifically vitreous body and retina. Their presence was justified by peripheral neutrophils that had infiltrated when BRB broke. In the same study, authors reported a reduction in NETs release after anti-VEGF therapy [113]. As shown, the research field on AMD is promising. 

## 12. Future Perspectives 

The overall aim of this review has been to condense current data of NETs related to healthy eyes and ocular diseases. As demonstrated, the better the understanding of molecular processes during inflammation, the better the assessment of intervention pathways to control it. 

Future work concerns taking a deeper approach of NET release mechanisms and molecules involved, in order to obtain new proposals that help with the diagnosis, prognosis and management of ocular pathologies. 

A better understanding of NETosis triggers during diseases may aid in discovering new therapeutic potential interventions based on ocular molecular components. Therapeutic interventions centred on clearing NETs as well as inhibiting their signalling molecules could yield innovative targets to work with. 

Molecule systems involved in NET release production should be a focus of future research. Assessment of those molecular changes implicated in the NET release process will provide new insights into the pathogenesis and therefore the potential management of a wide range of diseases. 

A schematic summary is presented in Figure 1. 

## Figures and Tables

**Figure 1 cells-08-00979-f001:**
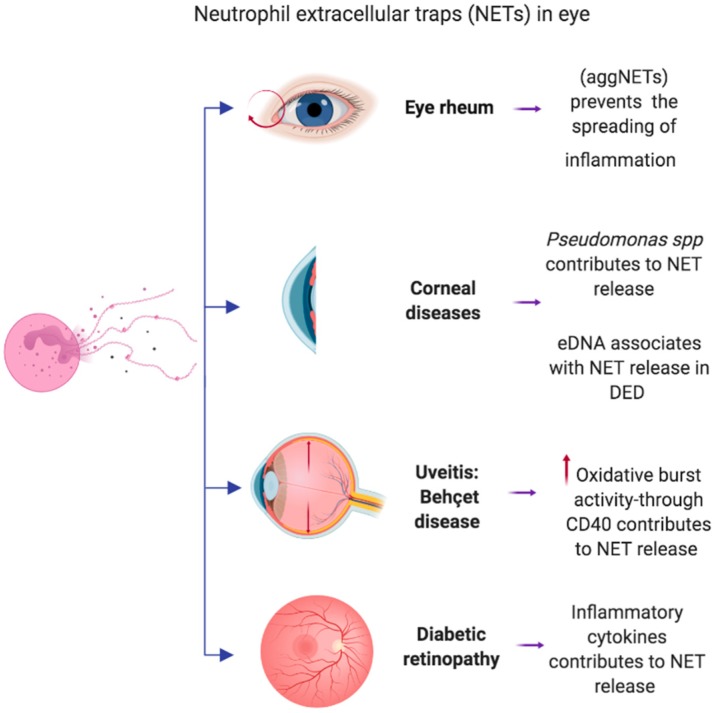
Neutrophil extracellular traps in eye. Neutrophil extracellular traps (NETs) are involved in several diseases in eye. NETs are associated with the pathophysiological mechanism, contributing to the exacerbation of the disease in some cases, such as diabetic retinopathy, in corneal diseases for instance, dry eye disease (DED), another disease is uveitis. However, a benefit of the NETs in eye, we can see it in the eye rheum, in which the NETs prevent the spread of inflammation. AggNETS: Aggregated neutrophil extracellular traps; eDNA: extracellular DNA; DED: Dry eye disease. Created with: biorender.com.

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
