# Peer review of "Neutrophil Extracellular Traps: Current Perspectives in the Eye"

_cells, 2019, doi:10.3390/cells8090979_

Round 1

Reviewer 1 Report

This manuscript reviews the involvement of NETs in the pathogenesis of eye diseases. It is a well written review and, based on reviewers knowledge, such a review is missing from the litterature.

Author Response

We appreciate the comments made by this reviewer. We have sent the entire manuscript to Scribendi Inc, which is an English editing service to amend grammar, syntax and typographical errors. 

Reviewer 2 Report

In this review article, the authors summarized the current knowledge about the prospective role played by NETs in ocular pathologies. The work is very interesting since, as far as I know, no review articles are available about this topic. The state of the art is extremely well organized, and includes a number of up-to-date information (about a quarter of the reports mentioned in this manuscript are published between 2018 and 2019).

However, there are some shortcomings that should be addressed before considering this manuscript suitable for publication:

1) The whole manuscript should be deeply and carefully revised with the support of an English editing service because of the presence of several grammar, syntax and typographical errors that sometimes heavily compromise the readability of this work. A language editing certificate should be provided.

2) Paragraph 7.1 "Meibomian gland dysfunction" should be removed from this review article, as it is inconsistent with the title and the scope of this work. Notably, no information is available on the role of NETs in meibomian gland dysfunction in this paragraph. Otherwise, the authors could integrate this paragraph by adding more specific information from the research article "Neutrophil extracellular trap (NETs) contribute to pathological changes of ocular graft-vs.-host disease (oGVHD) dry eye:implications for novel biomarkers and therapeutic strategies", Seungwon et al., 2019, The Ocular Surface. In this report, the authors highlight some involvement of NETs in meibomian gland dysfunction in oGVHD.

3) The following article should be clearly discussed in this review article, as it provides additional useful information about the role of NETs in confining ocular P. Aeruginosa biofilm, thus limiting the dissemination of bacteria in the brain. "Neutrophil extracellular traps confine Pseudomonas Aeruginosa ocular biofilms and restrict brain invasion", Thanabalasuriar et al., 2019; Cell Host Microbe.

Round 2

Reviewer 2 Report

As I previously mentioned, the authors made a great work with this very novel and interesting review article. In the new version of the manuscript, the authors properly addressed all the concerns raised by the reviewer and amended the manuscript accordingly. The revised version appears improved and easily readable.